# Embedded NiTi Wires for Improved Dynamic Thermomechanical Performance of Silicone Elastomers

**DOI:** 10.3390/ma13225076

**Published:** 2020-11-11

**Authors:** Umut D. Çakmak, Ingrid Graz, Richard Moser, Michael Fischlschweiger, Zoltán Major

**Affiliations:** 1Institute of Polymer Product Engineering, Johannes Kepler University Linz, Altenbergerstrasse 69, 4040 Linz, Austria; zoltan.major@jku.at; 2School of Education, Johannes Kepler University Linz, Altenbergerstrasse 69, 4040 Linz, Austria; ingrid.graz@jku.at; 3Soft Matter Physics, Johannes Kepler University Linz, Altenbergerstrasse 69, 4040 Linz, Austria; richard.moser84@gmail.com; 4Technical Thermodynamics and Energy Efficient Material Treatment, Clausthal University of Technology, Agricolastrasse 4, 38678 Clausthal-Zellerfeld, Germany; michael.fischlschweiger@tu-clausthal.de

**Keywords:** silicon elastomer, shape memory NiTi alloy, composite, dynamic thermomechanical analyses, damping, storage modulus, mechanical loss factor

## Abstract

The extraordinary properties of shape memory NiTi alloy are combined with the inherent viscoelastic behavior of a silicon elastomer. NiTi wires are incorporated in a silicon elastomer matrix. Benefits include features as electrical/thermal conductivity, reinforcement along with enhanced damping performance and flexibility. To gain more insight of this composite, a comprehensive dynamic thermomechanical analysis is performed and the temperature- as well as frequency-dependent storage modulus and the mechanical loss factor are obtained. The analyses are realized for the composite and single components. Moreover, the models to express the examined properties and their temperature along with the frequency dependencies are also presented.

## 1. Introduction

Ease of fabrication, transparency in the UV visible range, chemical inertness, low electrical conductivity, and their elasticity are some of the useful characteristics of Polydimethylsiloxane (PDMS) [1]. Therefore, PDMS, a silicon elastomer, is widely found in applications of fluidics, optical systems and sensors [2]. In addition, there is a growing interest in using PDMS in the field of robotics, as stretchable electronic skins [3], among many other applications. Martensitic phase transforming metals such as shape memory alloys (SMA) are, contrarily, hard inorganic materials (e.g., NiTi-alloys). Both materials have in common that they exhibit a nonlinear mechanical behavior. SMA’s behavior is also temperature, mechanical loading and in special cases magnetic field dependent [4]. These physical quantities induce a spontaneous change of the crystallographic lattice and lead to a change of the mechanical properties [4]. Due to their viscoelastic nature, PDMS’s mechanical behavior is time and temperature dependent. Both materials would exhibit a hysteretic behavior during mechanical cyclic excitation and, therefore, they are suitable for vibration damping [5,6].

In the literature various SMA hybrid composites were reported to achieve (large displacement) actuators [7,8,9], enhanced damping capacity as well as toughness within a thermoset polymer matrix [6,10], improved performance of self-healing polymers by embedding SMA wires [11,12], surface topography modulation [13], among others. Elastomer-SMA composites were of particular interest due to the combination of flexibility and shape memory effect initiated by the embedded shape memory alloy (long, short wires, plates, shells etc.). Applications were related to robotics and smart structures, as robot hands made of polyurethane-SMA composite [14] or dynamically programmable micro-wrinkles made of PDMS-SMA [13]. Moreover, in earlier investigations special emphases were given to actuator of any regard [15,16] and Huang (2002) [17] studied different SMAs for actuators and concluded that NiTi alloys revealed the best performance in terms of thermomechanical behavior. 

Motivated by the idea of merging the shape memory characteristic of NiTi and the soft flexible viscoelastic behavior of PDMS, we present a composite for enhanced damping capacity. Basically, PDMS is the matrix material and shape memory NiTi alloy is incorporated as wires. Figure 1 shows the barbell-shaped specimen made of PDMS with 4.4 vol% NiTi wires. Additionally, it illustrates the actuation capability of the composite. A mechanical deformation is temporarily preserved due to the metal wires and the flexible elastomer. When the temperature is elevated to the transition temperature, the SMA will regain the permanent shape resulting in the original straight shape of the sample. The temperature increase could be achieved by an external heating source or easily by electric heating through the SMA wires. Unlike shape memory NiTi wires, the temporary shape would never be preserved by the neat silicon elastomer without any external loading at ambient temperatures. 

Besides the actuation capability, also the damping performance of a PDMS can be enhanced by introducing SMA wires [5,6], however, it is less investigated compared to actuator designs. To demonstrate the damping capacity, a ball drop test is realized and shown in Figure 2. Two acrylic glass tubes were placed in a temperature chamber, where at the bottom of one tube the PDMS-SMA composite was fixed and the neat PDMS in the other tube for comparison purposes. The balls were held by electromagnets at the top of the tubes and by a trigger the balls could be released simultaneously. In Figure 2 the right tube belongs to the neat PDMS and the left one to the PDMS-SMA. A high-speed camera was utilized to record the ball drop. Sequences at four different temperatures of the first maximum rebound height are presented in Figure 2. Essential is the rebound height difference between the neat PDMS and the composite. From these observations it can be concluded that PDMS-SMA dissipates the energy more and faster; thereby it should be noted that the damping element made of PDMS-SMA is stiffer than that of neat PDMS of same volume, simply a result of the higher stiffness of the SMA.

To gain more insight into the interplay between PDMS and SMA, on the one hand, and the dynamic thermomechanical behavior, on the other hand, a comprehensive characterization methodology including an attempt to model the investigated mechanical properties and their temperature as well as loading rate dependency will be presented in the following. In this paper, we report the first study of combining NiTi wires with PDMS matrix, and investigate their stiffness as well as damping behaviors under dynamic thermomechanical loadings.

## 2. Experimental Section

### 2.1. Materials and Specimen

The silicon elastomer (PDMS, Sygard 184, DowCorning Inc., Midland, MI, USA) under investigation was mixed and cured with a base polymer:catalyst-ratio of 20:1. The NiTi alloy under investigation was a commercial round wire of Memry Corp. (Bethel, CT, USA). A superelastic composition (Ni_55.34_Ti_44.66_; indicated fractions in wt.%) was selected with a diameter of 760 µm. A thin diameter was preferred in order to have high compliance under compressive loading while the stiffness in tension was preserved. A volume content of 5% was selected to study the dynamic thermomechanical properties of PDMS/SMA composite. The SMA wire was investigated in the as received state with an oxide free surface finish and straight annealed.

For the experimental characterizations the “barbell”-shaped specimen was chosen in contrast to the standardized “dog-bone” specimen. This is caused by practical reasons and is mainly due to the advantages of reduced handling, gripping and alignment irregularities of specimens made of soft flexible materials as well as the higher forces achieved by the larger cross-section. Additionally, the barbell specimen can be utilized for uniaxial tension, compression and torsion loading. Here, we also preferred this specimen to rather have a single geometry to characterize the neat silicone elastomer and the PDMS-SMA composite, than have multiple specimens with their specific influences on the investigated mechanical properties. Since the same setup was utilized for the neat elastomer as well as the composite, the influences of the clamping and inertia effects were also the same. Figure 3 shows the dimensions of the barbell specimen. The material was cured in a custom-made cast mold in laboratory. For the composite, the NiTi wires were aligned simply with two plates with holes in the desired pattern at the top and bottom of the cast mold. Through these holes the wires were put and so misalignments during curing were prevented. The final composite is demonstrated in Figure 1.

### 2.2. Dynamic Thermomechanical Analysis (DTMA)

Figure 3 shows the experimental setup utilized to characterize the dynamic thermomechanical behavior under isothermal conditions at varying loading rates. Starting from the highest to the lowest, the investigated temperatures were set in decrements of 10 (5) K. At each isothermal temperature step the initial lengths due to the thermal expansion of the specimens were considered. Dynamic thermomechanical analyses (DTMA) were performed under sinusoidal uniaxial tension loading with dynamic displacement or stress amplitude and mean level displacement of stress. Prior to each DTMA of a specimen the electrodynamical actuator (TestBench, Bose Corp., ElectroForce Systems Group, Framingham, MN, USA) was tuned for a sine waveform in the desired amplitude and frequency range. The excitation frequency sweep covered a range with logarithmic steps (6 frequencies per decade) starting from the lowest first. A particular frequency was tested as long as a steady material response was achieved (i.e., creep and relaxation of the specimen was minimized). To characterize the influence of loading amplitude and mean level on the dynamic mechanical behavior of the neat PDMS as well as the PDMS-SMA composite, a respective dynamic mechanical test procedure at room temperature was conducted. This experiment was also of particular interest in order to demonstrate the benefits of the PDMS-SMA’s mechanical behavior compared to the neat matrix material. All experimental parameters are listed in Table 1. The neat PDMS as well as the PDMS-SMA composite were investigated with a displacement controlled procedure, while the NiTi alloy was examined in the stress controlled procedure. These varying control modes were chosen due to accuracy and limitation reasons. NiTi was tested in the form of a wire (small cross-section) and therefore the displacement amplitudes would have led to very high stresses and nonlinearities during mechanical excitation. Displacements were derived from the integrated LVDT (AD598, Analog Devices Inc., Cambridge, MA, USA) and forces were measured either with a 110-N (WMC-25lbf, Interface Inc., Atlanta, AZ, USA) or a 440-N (WMC-100lbf) load cell. WinTest DMA software (Version 7; Bose Corp.) was utilized to perform and analyze the DTMA experiments. 

The storage (E′) as well as the transient (E″ and tanδ) mechanical material properties were determined and can be expressed in the form of the complex modulus E* given by Equation (1). The well-known Prony series can be used to sufficiently model the dynamic material’s behavior in the frequency (ω = 2πf) domain [18]:(1)$$E^{*}\left(\omega \right)\;=\;E^{\prime }\left(\omega \right)+j\cdot E^{\prime \prime }\left(\omega \right)\;=E_{0}\cdot \left(1-\sum_{i}\frac{g_{i}}{1+{\left(\tau_{i}\cdot a_{T}\omega \right)}^{2}}+j\cdot \sum_{i}\frac{g_{i}\tau_{i}a_{T}\omega }{1+{\left(\tau_{i}\cdot a_{T}\omega \right)}^{2}}\right)$$
where E_0_ is the instantaneous modulus, τ_i_ the relaxation time and to this corresponding the Prony series coefficient is given by g_i_ = (E_i−1_ − E_i_)/E_0_. From Equation (1) the mechanical loss factor tanδ = E″/E′ can be directly derived:(2)$$\tan\delta \left(\omega \right)=\;\sum_{i}\frac{g_{i}\tau_{i}a_{T}\omega }{1+{\left(\tau_{i}\cdot a_{T}\omega \right)}^{2}-g_{i}}$$

If the material’s thermorheological behavior is simple, then the time–temperature superposition can be applied and a master curve determined for a particular reference temperature [19,20]. The frequency dependent mechanical characteristics’ curves (storage or loss moduli, Poisson’s ratio) at a temperature are simply shifted toward the curve progression of the reference temperature. For each isothermal curve a shift factor a_T_ is obtained and can be mathematically expressed by the well-known WLF-function [21]. The temperature-dependent shift factor a_T_ is given by:(3)$$\left|\log_{10}\left(a_{T}\right)\right|\;=\;17.44\cdot \frac{T-T_{g}}{51.6+T-T_{g}}$$

In an attempt to model the macroscopic temperature-dependent modulus of NiTi alloys, Cakmak et al. (2020) [22] proposed the following biphasic function:(4)$$E\left(T\right)\;={\;E}_{\mathrm{PT}}+\frac{E_{A}-E_{\mathrm{PT}}}{1+e^{\left(T-T_{0_{1}}\right)h_{1}}}+\frac{E_{M}-E_{\mathrm{PT}}}{1+e^{\left(T_{0_{2}}-T\right)h_{2}}}$$
where E_PT_ is the lowest modulus (phase transformation), E_A_ is the modulus of austenite dominant state of the alloy, E_M_ is the respective modulus of the martensite dominant state, T is the temperature, T0_1_ and T0_2_ are the temperatures related to the inflection points of the curves, and h_1_ and h_2_ are dimensionless constants of the exponential functions. This function will be utilized for the NiTi alloy under investigation. In addition, we desired to model the temperature- and frequency-dependent modulus parameter of the PDMS-SMA composite. Considering the viscoelastic behavior of PDMS by applying Prony series Equation (1) and the Equation (4) for NiTi alloys, the following coupled model was proposed:(5)$$E^{\prime }\left(T,\omega \right)\;=\;\left(E_{\mathrm{PT}}+\frac{E_{A}-E_{\mathrm{PT}}}{1+e^{\left(T-T_{0_{1}}\right)h_{1}}}+\frac{E_{M}-E_{\mathrm{PT}}}{1+e^{\left(T_{0_{2}}-T\right)h_{2}}}\right)\cdot \left(1-\sum_{i}\frac{g_{i}}{1+{\left(\tau_{i}\cdot \omega \right)}^{2}}\right)$$
where E′(T, ω) is the storage modulus of the PDMS-SMA composite. The instantaneous modulus E_0_ and its temperature dependency are represented by the biphasic function, while the frequency-dependent behavior is expressed by the real part of the Prony series. Equations (1)–(5) will be utilized to describe the measured and evaluated dynamic thermomechanical properties of the investigated materials and their correlations are also discussed in the following.

## 3. Results and Discussion

The fundamental characterization of the neat PDMS in terms of its viscoelastic behavior is presented in Figure 4. In the diagrams of Figure 4a–c, each color represents an isothermal measurement. The storage modulus E′ and the mechanical loss factor tanδ exhibit with increasing temperature a decline (cf. Figure 4a,b). Interestingly, the modulus measurements reveal that the temperature-dependency at higher frequencies is more pronounced while at 1 Hz it is almost not evident (see Figure 4a); the opposite is observable for the mechanical loss factor in Figure 4b. To determine the thermorheological behavior of the investigated PDMS, we have considered the wicket-plot in Figure 4c. This diagram is very convenient for analyzing the trend of the inherent respond timE′s temperature dependency as a result of the external mechanical loading. If all examined data reveal that tanδ is a unique function of E′, then the respond times (here the relaxation times) are equally temperature dependent and time–temperature superposition (TTSP) is applicable [19,23]. Here, we concluded that the measured data are with good approximation thermorheologically simple and TTSP can be applied. Moreover, from a generated master curve, shift factors can be determined and WLF-function (Equation (3)) could be applicable to model these factors. We constructed the master curve at the glass transition temperature Tg of the neat PDMS (157.15 K) and the determined shift factors revealed a temperature dependency, which is perfectly expressed by the WLF-function (see Figure 4d).

It is very convenient that the neat PDMS’s thermorheological behavior is simple, as the temperature- and rate-dependent mechanical behavior can be easily modeled. Moreover, its influence may be ascribable when the PDMS-SMA composite is characterized and in return we may get more insight into the effects of NiTi within the composite. The master curves of the neat PDMS’s storage modulus E′ and mechanical loss factor tanδ are illustrated in Figure 5a. Reference temperature therefor was the glass transition temperature of 157.15 K and it perfectly fitted the experimental data. The parameters of a 5th order Prony series were identified (see Table 2) for the modulus master curve and applied to tanδ according to Equation (2). In Figure 5a the predicted curves are illustrated with the respective correlation coefficients (R^2^) to the experimental data. At low and high frequencies the fit of the Prony series to the tanδ master curve is not as good as in the other frequency range. As the Prony series was optimized to the storage modulus, this misfit could be related to the earlier mentioned opposite temperature dependency of tanδ and E′ at low and high frequencies (cf. Figure 4a vs. Figure 4b). The macroscopic storage modulus and the tanδ of the NiTi alloy are shown in Figure 5b. In contrast to the earlier discussed neat PDMS, the time–temperature superposition principle is not applicable to the NiTi alloy. However, a pronounced temperature dependency can be observed for both dynamic mechanical properties. In addition, the mechanical loss factor seems to be loading frequency-dependent, while the storage modulus is almost independent. The varying mechanical properties are mainly caused by the diffusionless phase transformation of the microstructure from austenite to martensite passing through an intermediate phase (rhombohedral-phase). From low to high temperature, the storage modulus declines until a minimum of 12.5 GPa, followed by a steep increase to a rather plateau with small variations around 20 GPa. The tanδ behavior is opposite within the investigated temperatures, showing an increase until the peak is reached and declines subsequently. The peak of the martensitic phase transformation is measured at around the same temperature of 310 K as the minimum of the storage modulus. This seems to be reliable, as the damping is higher the lower the storage modulus is. From modeling point of view, we have not found any constitutive law to describe (predict) the macroscopic behavior of NiTi alloys in the literature. Therefore, some of the authors conducted an investigation focusing on the dynamic thermomechanical behavior of NiTi alloys [22]. The frequency discrepancy was highlighted and reported there, including an empirical model for the temperature-dependent modulus of NiTi alloys. It was found that the biphasic function according to Equation (4) is capable of predicting the storage modulus characteristic. The fit is demonstrated in Figure 5b as a dashed line and the determined model parameters are listed in Table 3.

The diagrams of Figure 4 and Figure 5 showed that the inherent viscoelastic characteristics of the silicon elastomer can be described by the well-known models for thermorheological simple behavior and the NiTi alloy’s temperature-dependent modulus can be modeled by an empirical proposed biphasic function. Now, the dynamic thermomechanical behavior of the PDMS-SMA composite will be discussed based on the evaluations summarized in the diagrams of Figure 6. The composite was evaluated and analyzed with a similar methodology as previously discussed. Wicket-plots (dissipative vs. storage properties) were considered to illustrate the loading-rate and temperature dependency of the PDMS-SMA. First of all, Figure 6a compares the neat PDMS performance with PDMS-SMA obtained by the amplitude and mean level sweep procedure at 10 Hz and room temperature. As expected the stiffness of the PDMS-SMA is higher, caused by the much higher modulus of the introduced NiTi wires into the PDMS matrix, while the volume of the investigated barbell-specimen is unchanged. However, it is interesting to note that tanδ is increased as well, meaning that the damping capability is also enhanced. Recalling the drop-ball test illustrated in Figure 2, we can conclude that the higher and faster dissipation of the PDMS-SMA damping element is a consequence of the increased stiffness as well as mechanical loss factor. The thermomechanical behavior of the composite measured at various frequencies is shown in Figure 6b. Each color represents an isothermal measurement. The frequency range is indicated by an arrow within the diagram. Generally, the higher the frequency, the higher is the modulus of the PDMS-SMA composite. To some extent this trend can also be observed for tanδ; although at low frequencies the mechanical loss factor is rather constant. An opposite observation is made for the temperature dependency of tanδ; the lower the temperature, the higher the value, however, at high temperatures and low frequencies the tanδ is approximately 0.15 and almost unaffected. While the tanδ is constantly increasing with decreasing temperature, the storage modulus reveals a decrease up to 35 °C (308.15 K) followed by an increase. Comparing this temperature with the transition temperature obtained from the diagram in Figure 5b, where the modulus of the NiTi alloy exhibits a minimum and tanδ a maximum, it follows that the PDMS-SMA’s modulus characteristic is dominated by the NiTi wires and shows also the same transition at around 310 K. On the other hand, the tanδ characteristic reveals no maximum at this temperature and, consequently, it must be dominated by the matrix material (PDMS). From this point of view, the observed frequency dependency is related to the PDMS matrix behavior, as the trend is similar to the earlier discussed and shown Figure 4a,b.

Moreover, at the boundary of the NiTi wires and the PDMS matrix is no perfectly bonding and so small relative movements between them cannot be prevented, leading to frictional dissipation of the external mechanical loading. A dissipative process is introduced by the non-perfectly bonded boundaries additionally to the bulk dissipation due to the viscoelastic behavior of the elastomer.

Focusing on the storage modulus, a 3D plot is presented in Figure 6c for the sake of better illustration of the compositE′s loading rate (frequency) and temperature dependencies (red data points). The 2D projected data for each parameter (blue and green) are also shown. Here, again the transition region of the modulus is visible and its frequency dependency.

In addition, a coupled model is utilized according to Equation (5) to predict the frequency- and temperature-dependent storage modulus characteristics. A 3rd order Prony series was combined with the proposed biphasic function. Thereby, the biphasic function was used to compute the instantaneous temperature-dependent modulus of the composite; the frequency dependency was covered by the Prony series. This approach is justified by the earlier argumentation, that the modulus and its temperature dependency are determined by the NiTi phase of the composite and the frequency dependency by the PDMS matrix. A correlation coefficient (R^2^) of 97.75% is achieved by this approach and the corresponding model parameters are summarized in Table 4. For the model parameter determination, the relaxation times were predefined in order to reduce the number of dependent variables and the well-known Levenberg-Marquardt algorithm was utilized. The evaluated standard errors for the derived parameters are presented in Table 4.

## 4. Conclusions

We demonstrated the actuation and damping capabilities of a composite made of a silicon elastomer as matrix material and NiTi alloy wires as filler component. The wires reinforce the matrix material and provide additional features, such as electrical conductivity along with thermal conductivity and the enhancement of the mechanical damping. In this research, we focused on the damping performance of the composite as well as the single components in dynamic thermomechanical experimental procedures. Besides these additional features, we discovered that with this simple composition the stiffness as well as the mechanical loss factor can be increased. Counterintuitive properties are achieved by the introduction of SMA wires leading to increased stiffness and hardness along with damping capacity. Furthermore, the properties can be tailored by the wirE′s specific alloy formulation. The temperature induced shape memory effect combined with the hyperelastic material behavior and the inherent viscoelasticity (time–temperature dependency) of the matrix PDMS, can be utilized to gain actuation capabilities (see Figure 1) and by exploitation of the Joule heating the matrix behavior can be varied (locally). The compositE′s stiffness is mainly dominated by the NiTi wires, while the mechanical loss factor is controlled by the matrix behavior. The attempt to model and predict the behavior of the composite confirmed, that the temperature dependency is primarily influenced by the instantaneous modulus parameter of NiTi (biphasic function) and the frequency dependency by the silicon elastomer (3rd order Prony series). The only unknown remains the macroscopic friction between the matrix and the wire due to the non-adhering interface. In future research this has to be improved and compared to the current observations.

## Figures and Tables

**Figure 1 materials-13-05076-f001:**
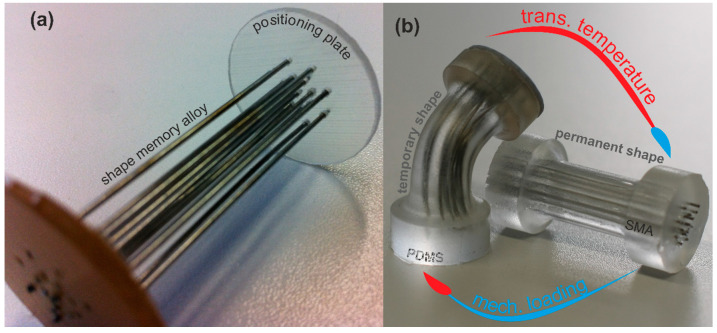
(**a**) Positioning plate with the shape memory NiTi alloy wires prepared for the cast mold of a barbell-shaped specimen; (**b**) Actuation principle of the barbell-shaped composite specimen of Polydimethylsiloxane (PDMS) (Sylgard 184, base polymer:crosslinking agent 20:1, DowCorning Inc., US) and 4.4 vol% NiTi (Memry GmbH, D).

**Figure 2 materials-13-05076-f002:**
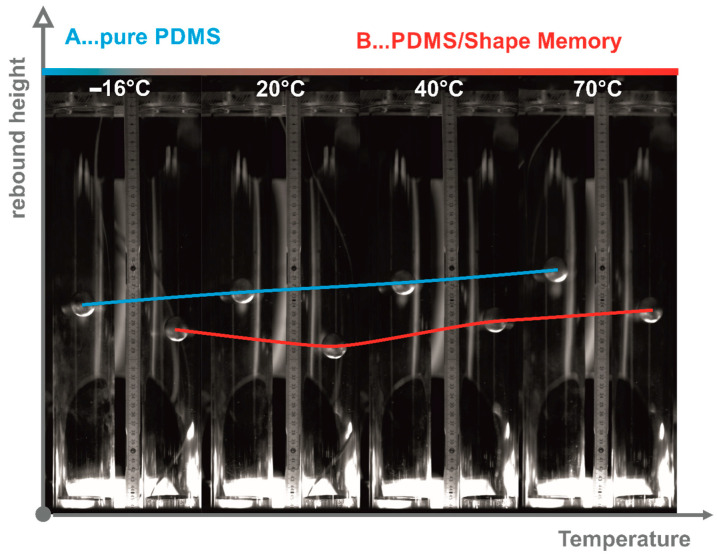
Ball drop test at various temperatures; **left:** neat PDMS; **right:** PDMS-SMA.

**Figure 3 materials-13-05076-f003:**
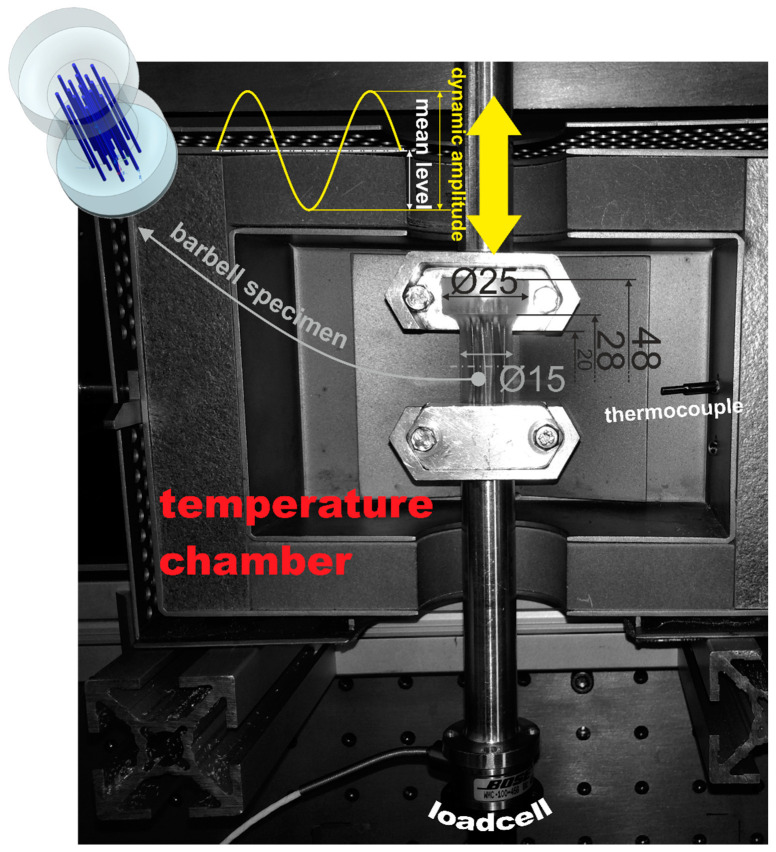
Experimental setup to characterize the dynamic thermomechanical behavior including the dimensions of the barbell specimen in mm.

**Figure 4 materials-13-05076-f004:**
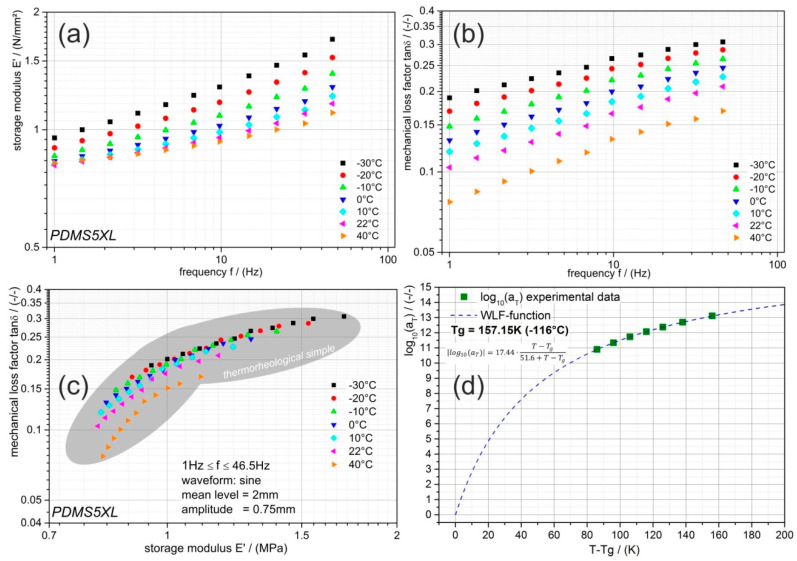
Temperature- and frequency-dependent dynamic thermomechanical behavior of neat PDMS. (**a**) frequency-dependent storage modulus E′ determined at various temperatures; (**b**) frequency-dependent mechanical loss factor tanδ determined at various temperatures; (**c**) wicket-plot; (**d**) temperature-dependent time–temperature shift factor a_T_ including the WLF-function (dashed line).

**Figure 5 materials-13-05076-f005:**
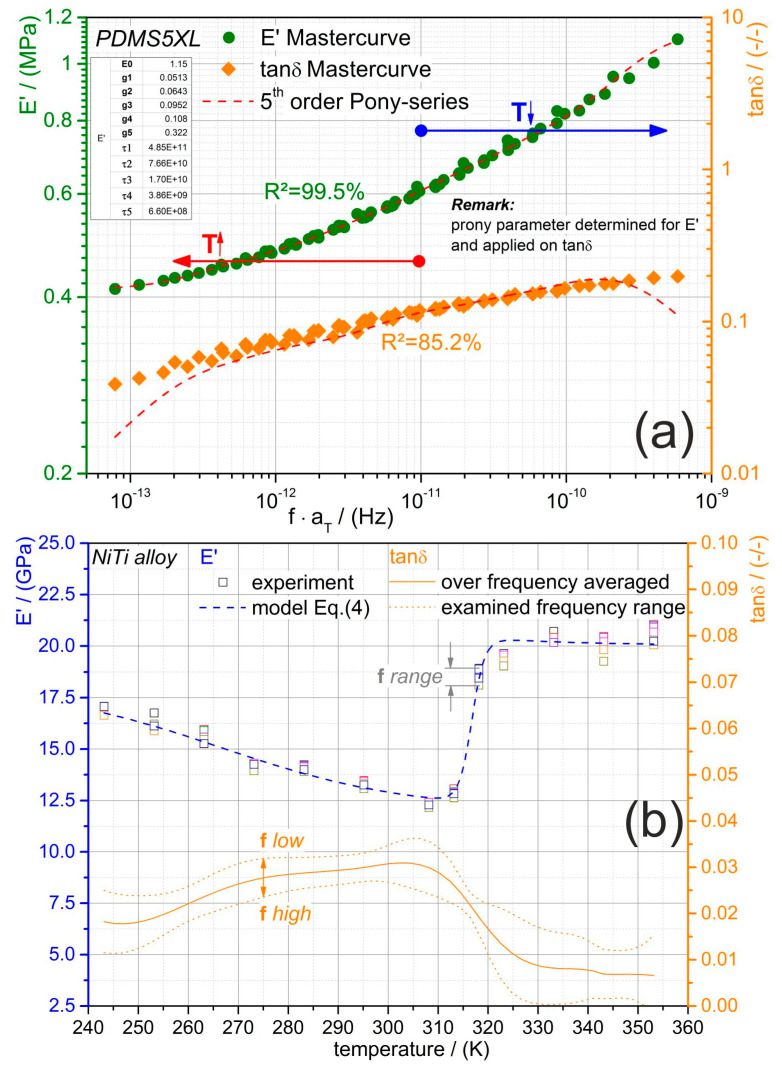
Modulus characteristics of the neat PDMS and the NiTi alloy determined by DTMA. (**a**) green data points show the master curve of neat PDMS at glass transition temperature for the storage modulus E′ and the orange data points show the corresponding master curve of the mechanical loss factor tanδ. The dashed lines represent the 5th order Prony series (Equations (1) and (2)) prediction with the parameters shown in the inserted table. (**b**) temperature-dependent storage modulus as well as mechanical loss factor tanδ characteristic of the NiTi alloy. Dashed blue line is the temperature dependent modulus modeled by Equation (4). The full orange line is the over the frequency averaged tanδ curve; the dashed orange lines indicate the maximum as well as the minimum tanδ trend.

**Figure 6 materials-13-05076-f006:**
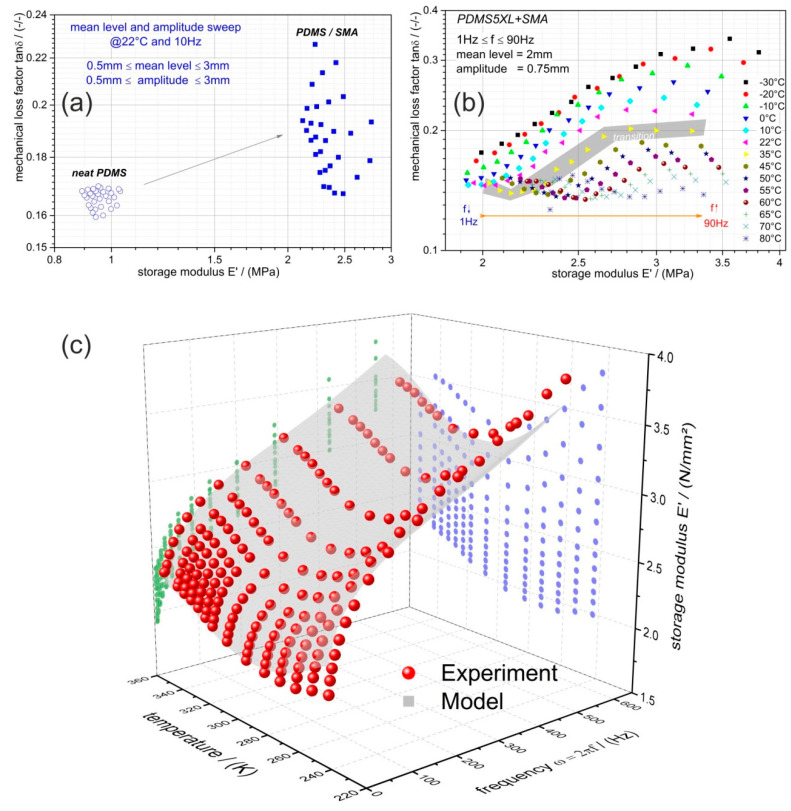
Illustration of the DTMA results and performance of PDMS-SMA composite. (**a**) The mean level as well as dynamic amplitude sweep data of neat PDMS (circle) in comparison with PDMS-SMA (square) depicted in the wicket-plot. (**b**) The temperature- as well as frequency-dependent results of PDMS-SMA. (**c**) 3D-plot of the temperature- and frequency-dependent storage modulus E′ of PDMS-SMA (red points); blue data points…projection to E′ vs. temperature; green data points…projection to E′ vs. frequency; grey area…prediction according to Equation (5).

**Table 1 materials-13-05076-t001:** Experimental parameters for the investigated materials during dynamic thermomechanical analysis (DTMA).

Material	Dyn. Ampl.	Mean Level	Frequency	Temperature
PDMS5XL	0.75 mm	2 mm	1 Hz–46.5 Hz	−30 °C–+40 °C
	0.5 mm–3 mm	0.5 mm–3 mm	10 Hz	+22 °C
NiTi	8 MPa	25 MPa	1 Hz–21.5 Hz	−30 °C–+80 °C
PDMSSMA	0.75 mm	2 mm	1 Hz–90 Hz	−30 °C–+80 °C
	0.5 mm–3 mm	0.5 mm–3 mm	10 Hz	+22 °C

**Table 2 materials-13-05076-t002:** Model parameters of the 5th order Prony series according to Equation (1) for PDMS.

E_0_	g_1_	g_2_	g_3_	g_4_	g_5_	τ_1_	τ_2_	τ_3_	τ_4_	τ_5_
MPa	-	-	-	-	-	s	s	s	s	s
15	0.0513	0.0642	0.0952	0.108	0.322	4.85 × 10^−11^	7.66 × 10^−10^	1.70 × 10^−10^	3.86 × 10^−9^	6.60 × 10^−8^

**Table 3 materials-13-05076-t003:** Model parameters of the biphasic function according to Equation (4) for NiTi alloy.

E_PT_	E_A_	E_M_	T0_1_	T0_2_	h_1_	h_2_
MPa	MPa	MPa	K	K	-	-
12,020	17,810	19,657	266.55	315.75	0.06	0.87

**Table 4 materials-13-05076-t004:** Model parameters of the biphasic function coupled with the 3rd order Prony series according to Equation (5) for the PDMS-SMA composite.

Parameter	Unit	Value	Standard Error
E_PT_	MPa	7.248	0.3033
E_A_	MPa	8.771	0.7537
E_M_	MPa	8.696	0.3047
T0_1_	K	253.135	10.0159
T0_2_	K	326.720	3.3931
h_1_	-	0.083	0.0541
h_2_	-	0.078	0.0266
g_1_	-	0.045	0.0031
g_2_	-	0.055	0.0051
g_3_	-	0.642	0.0119
τ_1_	s	0.063	predefined
τ_2_	s	0.009	predefined
τ_3_	s	0.001	predefined
R^2^	%	97.75

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
