# Peer review of "Embedded NiTi Wires for Improved Dynamic Thermomechanical Performance of Silicone Elastomers"

_materials, 2020, doi:10.3390/ma13225076_

Round 1

Reviewer 2 Report

The authors demonstrated the actuation and damping capabilities of a composite made of a silicon elastomer as matrix material and NiTi alloy wires as filler component. 

The letters of the all numerical formulas are garbled. Therefore it has some difficult to read this article. The right letters, please.

In table 4, the evaluated standard errors for the derived parameters were listed. The errors of  the parameters of EA, h1, hwere not small for the obtained values. Explain the proof of these obtained values being appropriate.

Revice this article according to mdpi materials format (Chapter setting, reference lists).

Please see the attachment for other comments
